# Technology Spillover Effect in China: The Spatiotemporal Evolution and Its Drivers

**Chengliang Liu [1,2,3] and Qingbin Guo [4,*]**

[1]    Institute for Global Innovation and Development, East China Normal University, Shanghai 200062, China; clliu@re.ecnu.edu.cn

[2]    School of Urban and Regional Sciences, East China Normal University, Shanghai 200241, China

[3]    Institute of Eco-Chongming, East China Normal University, Shanghai 200062, China

[4]    School of Economics, Hainan University, Haikou 570228, China

[*]    Correspondence: gqbhust@aliyun.com

**Abstract:** Under the background of global economic integration, the technology spillover effect is playing a more and more important role in the technological progress of developing countries. In this circumstance, this paper conducted an in-depth analysis on the 12-year spatial-temporal evolution of the international technology spillover effect and its driving determinants in China during the period of 2003 to 2014. Analytical results highlighted that: (1) As a whole, the international technology spillover effect in general has shown an upward trend in China, except in 2008 and 2012, which are observed as deep-V plunge variations. The plunge in 2008 was more dramatic. After 2011, the growth rate of international technology spillover effect from international import trade and foreign direct investment (FDI) respects slowed obviously down. (2) The spatial distribution of the international technology spillover effect from FDI in China transferred from the ribbon-like pattern to the flake-like pattern, while the effect from import trade held steady with little difference in regional spatial distribution. (3) From the drivers, human capital, economic development, trade openness, and institutional factors promoted the technological spillover effect of import trade channels positively, and financial development, human capital, economic development, and institutional factors promoted the technological spillover effect of FDI channels positively.

**Keywords:** technology spillover effect; spatial-temporal evolution; foreign direct investment (FDI); import trade

## 1. Introduction

To date, capital accumulation and technological advancement are the two main driving forces of economic growth. Among them, technological advancement is even considered as an internal power of the long-term economic growth by endogenous growth theories. In the knowledge-based era, technological progress plays an increasingly important role in the economic growth of developing countries, in which technological advances depend on their research and development (R&D) capability and absorption of all possible technology spillovers. Generally speaking, the proportion of R&D in gross domestic product (GDP) in developing countries is far less than that in developed countries. Therefore, the technological advances of developing countries mainly rely on the absorption of technology spillovers from developed countries.

International technology spillover refers to the externalities that foreign research and development activities exert on their own countries through a verity of commercial activities, such as international trade and investment. Therefore, there are mainly two channels for technology spillover. One is brought by FDI, that is, multinational corporations take use of their technological and capital advantages in the

home country to set up or merge enterprises in the host country, which will promote knowledge and technology spillover. The other is international import trade that also can bring about international spillovers of technological and knowledge information.

FDI has been viewed as a pillar of China's economic growth since the reform and opening up. A number of empirical studies have proved that FDI has indeed promoted China's economic growth. On the one hand, the inflow of FDI brought exceptional benefits, including domestic economic growth, technological advances, and industrial restructuring. On the other hand, these benefits led to certain predicaments, such as environmental issues and a lack of core technology. What is more, China still occupies a middle or lower position in the international trade value chain. The product life-cycle theory suggests that, because China cannot imitate or assimilate one technique until it attains maturity, its technology spillover effect is bound to gradually decrease. In addition, a country's technological growth relies not only on technology spillover effect, but also on its human capital and research and development (R&D).

At present, few scholars analyze the effect of international technology spillover from the geographical perspective. In view of the current situation, the paper analyzes the spatial-temporal evolution of the international technology spillover effect and its drivers. The change of the effect of international technology spillover in China will be introduced in chronological order and the distribution of international technology spillover in space will be shown in the paper.

China has nagged by the unbalanced development of regional economy for a long time. For example, the labor productivity in the Eastern coastal areas is higher than that in the central and Western regions. We explore the distribution of the stock of international technology spillover in time and space to the introduction of foreign capital and the optimization of its spatial pattern in a better and reasonable way. Up to now, China has finished the reform and opening-up in a relatively short period of time and basically realized an overall well-to-do society. There is, however, still a striking gap between the rich and the poor in China. From 1995 to 2014, the top ten provinces and cities with per capita GDP rankings are basically the Eastern coastal areas, such as Beijing, Shanghai, Zhejiang, Guangdong, and Tianjin, while the last five are basically in the Western region. Overall, the per capita GDP of the top ten provinces and cities is far higher than that of the central and Western regions. The difference in per capita GDP between the central and Western regions is small. The aim of our empirical exercise is to study the distribution and determinants of international technology spillovers are also beneficial to contain the gap between the rich and narrow the gap among regions.

## 2. Literature Review

"International technology spillover effect" has been a hot issue for a long time. Coe and Helpman [1] are pioneers to measure the term through international trade. By focusing on a sample of 21 OECD members and Israel over the period 1971–1990 and measuring the international technology spillover effect by using the import-shares index, they found that a positive correlation between R&D of the domestic trade partners and the domestic Total Factor Productivity (TFP). What is more, the higher degree of trade openness, the greater the impact on the country. In 1998, Lichtenberg and Van Pottelsberghe [2] improved the method of measuring international technology spillovers of import trade channels by using the index of the proportion of import-shares in GDP. Meanwhile, they also put forward the measure of technology spillover effect from the FDI channel.

Subsequently, researchers have carried out a large number of empirical studies based on the CH (Coe and Helpman) model or the LP (Lichtenberg and Van Pottelsberghe) model. Most existing studies have confirmed that import trade and FDI are two important channels of international technology spillover effect [3]. In fact, these technology spillover channels not only include FDI and import trade, but also export trade, outward direct investment, immigration effects, international journals, academic conferences, patent licensing, and industrial espionage, among others. However, research in these fields is relatively limited [4].

More studies support import trade as the main technology spillover channel and note that the positive effect of import trade on international technology spillovers is stable. Based on G7 data, Keller [5] concluded that international trade benefits international technology spillovers and increases the domestic total factor productivity (TFP). Based on the CGE model, Parrado and Enrica [6] validated a material, outstanding international spillover effect on foreign trade channel.

Similar to the technology spillover effect through import trade, many scholars believe that FDI has positive technology spillovers. For example, when studying the technology spillover effect of FDI in Iran, Salim et al. [7] found that the technological capability of subsidiary corporations, working as intermediaries, exercised a positive influence through two spillover channels: The demonstration effect and training effectiveness. Djulius [8], who explored a possible increase in domestic worker productivity related to workforce availability at foreign firms, found that domestic firms' export orientation can promote a spillover of knowledge. Liu et al. [9] believed that, through FDI, technological innovation could be incentivized by FDI and other sources, such as R&D activities, physical and human capital, and regional technology spillovers. Other scholars believed that the international technology spillovers through FDI are not distinct or uncertain. Huang and Zhang [10] thought that FDI crowded out domestic enterprises. Naveed and Ahmad [11] found that technology and knowledge spillovers from the European Union (EU) derive mainly from neighboring regions, that spillover effect across the international borders are statistically insignificant, and that inaccurate estimates of spillovers are caused by a border effect. Girma [12] believed that a minimum absorptive capacity threshold level exists, below which productivity spillovers from FDI are negligible or even negative. Kwon and Chun [13] found that when local companies lag far behind multinational corporations in technical skill, though technology transfer diffusion, and the former are able to absorb technology, and a technology spillover may not arise. Ubeda and Perez-Hernandez [14] showed, with threshold regressions, that a non-linear relationship between FDI and productivity improvement in domestic firms, which is conditioned by absorptive capacity and geographical distance. Firms with high absorption capacities benefit from positive spillovers, while other firms are negatively affected by the presence of multinational corporations.

In summary, most existing studies have confirmed that while import trade and FDI are two important channels of international technology spillover effect, few scholars analyze the spatial-temporal evolution of the international technology spillover effect, and so are different from the research of Zhang et al. [15], who analyzed the influencing factors, regional differences, and temporal variations of technology innovation, this paper discusses the spatial-temporal evolution of the international technology spillover effect from import trade and FDI channels and its drivers, and policy recommendations, based on spatial-temporal evolution and its determinants are more targeted, which will help China to make better utilization of the technology spillover effect. In addition, it can guide developing countries to undertake and absorb technological spillovers from developed countries through the study of the temporal and spatial evolution of international technology spillover effect in this paper. Thereby, they will enhance the technological innovation capabilities of developing countries and their position in global value chains, promote the transformation of traditional development modes, and achieve sustainable economic and social development.

## 3. Method and Data

Concerning the availability of data, the time-series samples are selected from 2003 to 2014, while the cross-section samples are selected from 30 provinces, autonomous regions, and municipalities (excluding Tibet) in mainland China. In addition, these provinces are divided into three regions, Eastern, central and Western, in order to form the panel data to show panel data. And the provinces are merged into three in Eastern, central and Western regions. The Eastern region includes 11 provinces and cities such as Beijing, Tianjin, Shanghai, Shandong, Hebei, Liaoning, Jiangsu, Zhejiang, Fujian, Guangdong, and Hainan. The central region includes Hubei, Hunan, Jilin, Heilongjiang, Anhui, Jiangxi, Shanxi, and Henan Eight provinces; the Western region includes Inner Mongolia, Shaanxi,

Sichuan, Chongqing, Guizhou, Guangxi, Yunnan, Gansu, Qinghai, Ningxia, and Xinjiang 11 provinces and autonomous regions, in order to form the panel data. Below this article on the foreign technology spillover stocks and the spatial distribution of our provinces and cities for the following description. In addition, these provinces are divided into three regions, Eastern, central and Western, to show panel data. Among them, the Eastern regions consist of 11 provinces, namely Beijing, Tianjin, Shanghai, Shandong, Hebei, Liaoning, Jiangsu, Zhejiang, Fujian, Guangdong, and Hainan; the central regions cover eight provinces, namely Hubei, Hunan, Jilin, Heilongjiang, Anhui, Jiangxi, Shanxi, and Henan; the Western regions include 11 provinces, districts and cities, namely Nei Monggol, Shaanxi, Sichuan, Chongqing, Guizhou, Guangxi, Yunnan, Gansu, Qinghai, Ningxia, and Xinjiang. The stock of international technology spillovers and its spatial distribution in different provinces will be introduced in the following parts:

### 3.1. The Measurement of Technology Spillover Effect by Import Trade Channel

Coe and Helpman [1] are pioneers who take the import-share weighted R&D as the index to estimate the stock of the international technology spillovers from international trade. Their equation has the following specification:

$$IMSP\text{it} = \sum_{j \neq i} \frac{Mijt}{Mit} SDjt \tag{1}$$

where $IMSP_{it}$ is the volume of foreign R&D capital stock spillovers from import trade of country (or region) $i$ in the year $t$, $M_{ijt}$ are the imports of country $i$ from country $j$ in the year $t$, and $M_{it}$ are the imports of country $i$ from other countries or regions in the year $t$. $SD_{jt}$ is the domestically produced R&D stock of country $j$ in the year $t$.

Later, Lichtenberg and Pottelsberghe [2] improved the calculating method of the international technology spillovers from import trade. The spillovers are expressed by the proportion of import in GDP rather than import shares in their opinion. The equation is written as follows:

$$IMSP\text{it} = \sum_{j \neq i} \frac{Mijt}{Yjt} SDjt \tag{2}$$

Here $Y_{jt}$ stands for GDP of country $j$ in the year $t$. The meaning of other variables refers to Equation (1).

### 3.2. The Measurement of Technology Spillover Effect by FDI Channel

Lichtenberg and Pottelsberghe [2] put forward the measurement index for the foreign technology spillover from FDI at the same time. It is calculated by the following equation:

$$FDISP\text{it} = \sum_{j \neq i} \frac{Fijt}{Kjt} SDjt \tag{3}$$

where $FDISP_{it}$ represents spillover volume of foreign R&D capital stock from FDI of country $i$ in the year $t$, $F_{ijt}$ is the direct investment from $j$ to $i$ in the year $t$, $K_{jt}$ denotes gross fixed capital formation of country $j$ in the year $t$ and $SD_{jt}$ stands for the domestic stock of R&D capital in country $j$ in the year $t$. This calculating method developed by Lichtenberg and Pottelsberghe [2] is widely applied by the subsequent scholars, including Keller [5], Edmond [16], Comin and Hobijn [17], Madsen [18], and so on. After comparing the two indexes, Li and Zhu [19] tend to adopt the one proposed by Lichtenberg and Pottelsberghe [2]. The measurement index of spillover used in the paper also follow their calculating method.

According to the main source countries of import trade and FDI in China and the R&D stock volume in each country, the paper selects the Group of Seven (G7, namely America, German, Britain, France, Japan, Canada and Italy), Hongkong, Korea and Singapore as research samples with a focus on

China's technology spillover effect from the import and FDI among G7 nations, Hongkong, Korea, and Singapore. The data on the gross fixed capital formation of these 10 countries or regions is obtained from World Investment Report 2015; the data on GDP of these countries come from the WDI database and all the related data have undergone conversion according to the constant price index in 2003; the data on R&D stock in the first year 2003 come from Guo and Fang [3], the expenditure on foreign R&D after 2003 is calculated by the equation that the ratio of R&D expenditure to GDP published by OECD Factbook 2015–2016 multiply by GDP. The data on R&D stock from 2003 to 2014 are calculated based on the perpetual inventory method.

## 4. The Temporal Dynamics of China's Technology Spillover Effect

### 4.1. The Temporal Variations of Technology Spillover Effect through Import Trade

The international technology spillovers through import trade presented rising trend within the given time range as a whole. However, there are several key turning points, such as the beginning of a decline in 2008 and the downturn in 2009, followed by a rapid rise from 2009 to 2011, and a gradual upward trend after 2011 (Figure 1).

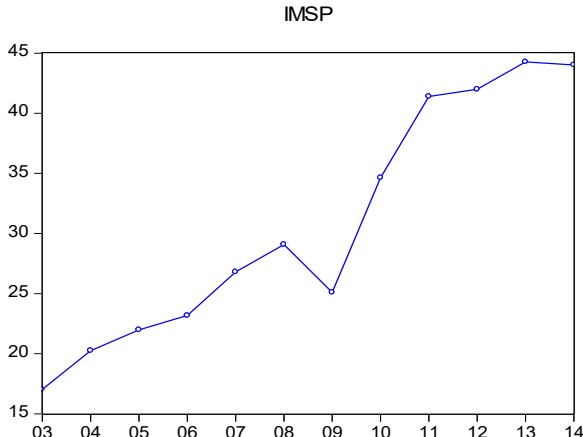

**Figure 1.** The temporal variations of China's technology spillover effect through import trade.

Overall, the international technology through international trade played a positive role in China. However, the decline of the international technology spillover effect in 2008 was mainly due to the financial crisis and economic crisis caused by the American subprime mortgage crisis swept across the globe in 2008 which also affected China. China's financial and economic conditions have also been hit to a certain degree. In particular, China has become the second largest in the world in the economy, the import ratio has been increasing year by year. The crisis has had a certain impact on China's import trade.

Under the influence of global financial crisis, two momentous changes, the re-balancing of global economy and the imposition of carbon tariff, have affected and will continue to affect China's development of import trade during the post-crisis era. On the one hand, they have promoted the proportion of import in trade, thus improving the importance of Chinese market in the international community; on the other hand, they call for and provide the direction for China's reconstruction of economic structure and upgrading of industrial structure. After the crisis, from 2009 to 2011, with the growth of import trade, the international technology spillovers through import trade picked up promptly. Since 2011, the growth rate of international spillovers began to slow down, mainly due to the policy adjustment for import and export. Before 2011, China had been emphasizing the "bring in" strategy. However, after 2011, China's economy started to transfer from the "breakneck growth" model to the "medium-high growth" model, which was accompanied by several problems, such as the rising downward pressure on economy and excess production capacity. In line with the status quo,

it implemented the international trade policy, which attached equal importance to "go global" and "bring in" strategies. In this way, it also could obtain the reserved international technology spillovers through export trade.

The paper estimates the effect of international technology spillover mainly on the basis of data on import trade, without considering the export trade. Nevertheless, actually, China's export volume has substantially gone up in recent years with the increasing international technology spillover into it.

### 4.2. The Temporal Variations of Technology Spillover Effect through FDI

Within the given time span, international technology through the transmission mechanism of FDI had a positive effect on China. The international technology spillover from FDI reflected an uptrend as a whole. However, in 2005, 2009, and 2012, there was a certain decline and then a rise in the V-shaped trend (Figure 2).

The international technology spillover from FDI has a close relationship with the amount of FDI. FDI increased over time in an overall view, but in 2005, 2009, and 2012, there were varying degrees of decline. According to the statistics bulletin, the actual utilized amount of FDI in 2005 was 60.3 billion dollars, a drop of 0.5%. In 2009, the actually used amount was 90 billion dollars, down by 2.66% over the same period of last year. In 2012, the total amount of FDI reached 111.7 billion dollars, down 3.7%. Therefore, it is believed that the decrease in the international technology spillover effect of FDI in 2005, 2009, and 2012 has something to do with the reduction of FDI.

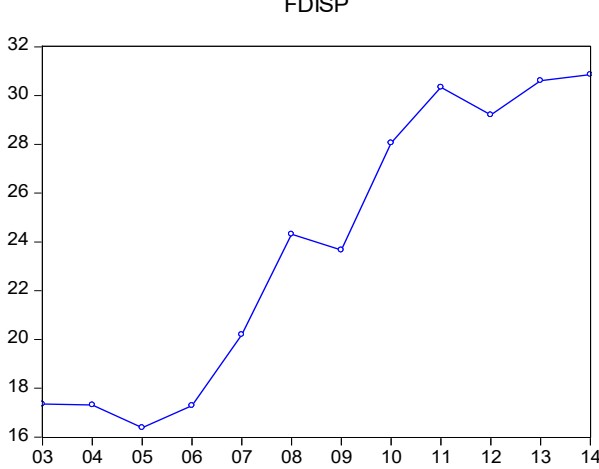

**Figure 2.** The temporal variation of China's technology spillover effect through FDI.

The V-shaped curve in 2009 resulted from the economic recession caused by the global financial crisis in 2008. During that period, the developed countries that were experiencing economic downturn would tend to reduce their investment in developing countries and regulate the macro economy at home, trying to revive their economy. For this reason, the international technology spillover that China receives will decline somewhat in 2009.

There are three possible reasons for the emergence of the v-shaped pattern in 2012. First, the growth range of FDI into China from developed countries or regions was insignificant. According to the National Bureau of Statistics, the increase of the actual amount of FDI received by China was not obvious from 116.011 billion dollars in 2011 to 119.562 billion dollars in 2014. Second, the variation in the gross fixed capital formation from the developed countries, which were sources of technology spillovers, was small. Third, the stock of the domestic R&D capital in developed countries or regions remained flat. The tendency also reflects the policy change of part of developed countries in outward foreign direct investment (OFDI). FDI is made through multinational corporations. For them, profit maximization is their ultimate purpose. In consideration of multiple factors, including policy, labor cost, and security, China is the best choice among developing countries for them to make FDI.

Nevertheless, with the constant improvement of the national economic strength and labor productivity, based on microeconomic theories, the labor cost borne by transnational corporations has been increasing. Moreover, the gradually reduced cheap labor made China lose its advantages in attracting FDI, while the massive and cheap workforce from Southeast Asian countries, such as Thailand, Cambodia, and Laos, were likely to capture the direct investment of transnational corporations. The rising cost resulting from the increasing compensation for employees and currency appreciation promoted transnational corporations to expand their business to other regions. Since 2012, a few multinational enterprises have begun to withdraw capital from China and set up factories in the countries which enjoyed cheaper labor, including several famous manufacturers of shoes like Adidas and Nike. In addition, after the sweeping economic crisis in 2008, developed countries have seen economic resurgence, part of which has went through the economic depression and made the policies of returning manufacturing industry to the home country. It would also affect the decision-making of multinational enterprises. For the above-mentioned reasons, China was partly influenced in the acquisition of international technology spillovers with V-shaped patterns.

## 5. The Spatial Dynamics of China's Technology Spillover Effect

### 5.1. Evolutionary Technology Spillover Effect through Import Trade

During the period 2003–2014, the evolution of spatial difference of the international technology spillover from import trade held steady with little difference in regional spatial distribution (Figure 3). The paper divides the international spillover effect into five levels: High spillover effect, relatively high spillover effect, medium spillover effect, relatively low spillover effect, and low effect. In general, the spatial distribution pattern of international technology spillover effect of import trade channels is relatively scattered and fragmented. The regions that presented high international technology spillover effect from import trade were Shandong, Jiangsu and Guangdong. Most of the regions that keep the international technology spillover effect from import trade at a relatively high level were around these regions that presented high spillover effect, including Hebei, Henan, Zhejiang, and Nei Monggol that reached the relatively high level from the medium level in 2006 and Shanghai that fell from the relatively high level to the medium level. The regions whose international technology spillovers were at the medium level were mainly distributed in the southern and northeast, such as Hubei, Anhui, Hunan, Heilongjiang, Liaoning. The effect of the international technology spillovers from import trade in the western region maintained a relatively low level, such as Xinjiang, Qinghai, Yunnan, Guizhou, and Guangxi.

In the period of 2003–2014, the spatial distribution pattern of the international technology spillover from international trade basically remained stable, but some provinces and cities also had the spillover of rising or falling. For example, the spillover effect of Nei Monggol rose from the medium level to the relatively high level, whereas the effect of Beijing decreased from the medium level to the relatively low level and that of Shanghai fell from the relatively high level to the medium level. Furthermore, the developed regions, including Beijing, Tianjin and Shanghai didn't present obvious technology spillover effect. The two main reasons have been put forward in the paper. First, these regions did not have many import shares, while some provinces like Guangdong whose manufacturing are flourishing enjoyed plenty of imports. Second, international technology spillovers arise from the technological disparity between the host country and the home country, the wider the gap in technology between the two sides is, the more spillovers the host country will receive. However, as these developed areas enjoyed a great number of talents and technologies and led domestic R&D, the technology spillovers from import trade into them was relatively limited.

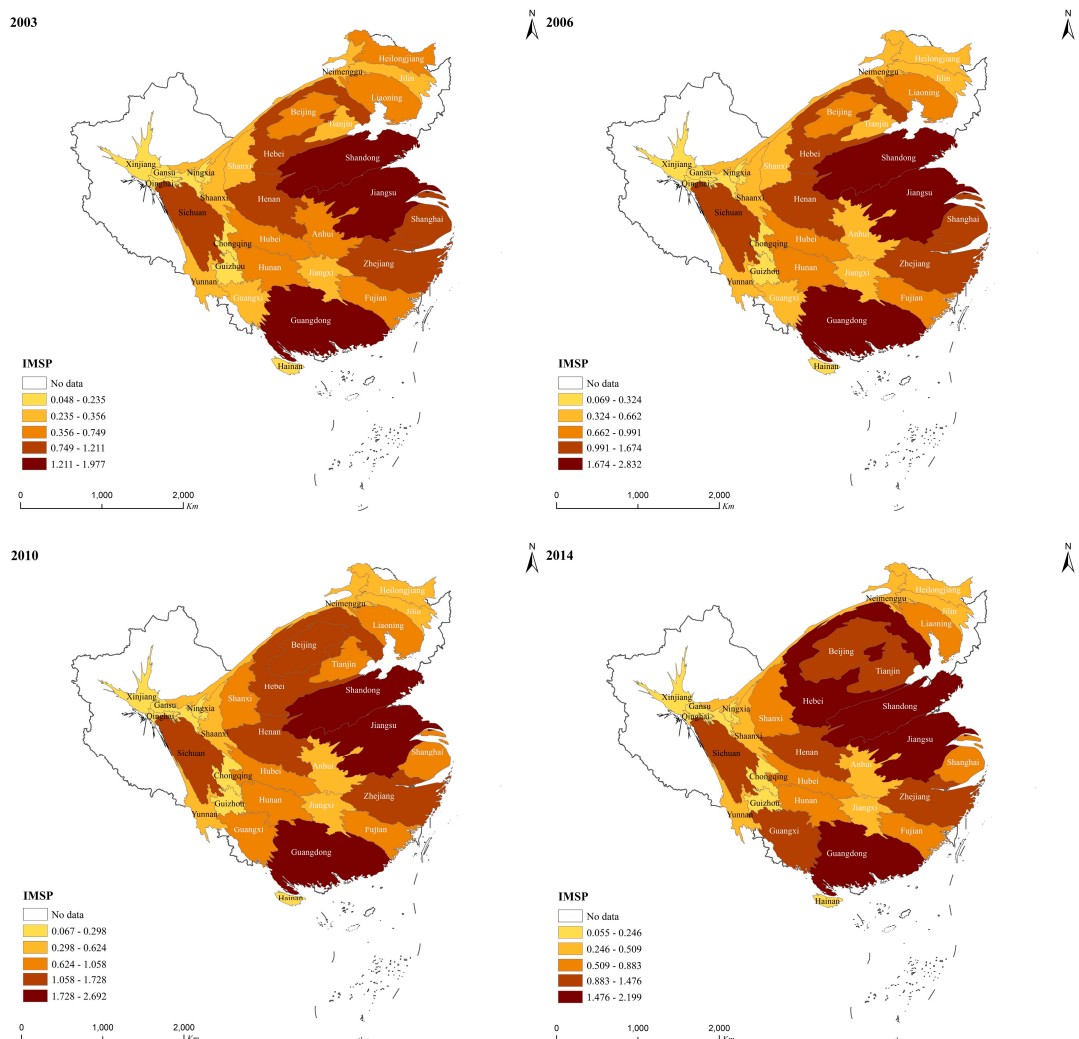

**Figure 3.** The spatial dynamics of China's technology spillover effect through import trade.

*5.2. Evolutionary Technology Spillover Effect through FDI*

As shown in Figure 4, the spatial distribution of the international technology spillovers from FDI in China transferred from the ribbon-like pattern to the flake-like pattern from 2003 to 2014. The paper divides the effect of the international technology spillovers in the channel of FDI into five levels: High spillover effect, relatively high spillover effect, medium spillover effect, relatively low spillover effect, and low effect. In 2003 and 2006, the high-level spillover effect covered the economically developed provinces along the eastern coast, including Shandong, Jiangsu, Zhejiang, and Guangdong and the ribbon-like distribution feature in these areas was quite obvious. The regions that had relatively high spillover effect were some coastal cities, such as Shanghai and Fujian, as well as provinces in central China, including Hebei, Henan, Hubei, and Hunan. The regions that had a medium spillover effect included Northeast China, Anhui, Jiangxi, Shaanxi, Yunnan, and Guangxi. The relatively low spillover effect and low effect are mainly concentrated in the Western region.

In 2010, the levels of international technology spillover effect from FDI in some provinces reduced, breaking the ribbon-like pattern. For example, Zhejiang declined from the high level to the relatively level and Hunan from the relatively level to the medium level. The areas generating the high spillover gradually shrank, while the medium effect continued to expand, forming the agglomeration pattern of the spillover effect from FDI in space. In 2014, the relatively high effect of regions further widened. For example, Hunan and Anhui rose from the medium level to the relatively high level. By 2014, except Jiangxi, the spillover effect of the coastal and central regions was at the relatively high level or above.

In addition, in the statistical period of 2003–2014, there is no significant change in the distribution of low spillover effect from FDI in the range from 2003 to 2014, mainly in the Western region. While the spillover effect in the Western region continues to rise, overall, the spatial distribution pattern of the low-level spillover effect remains basically unchanged, while the Western region is still in a weak position.

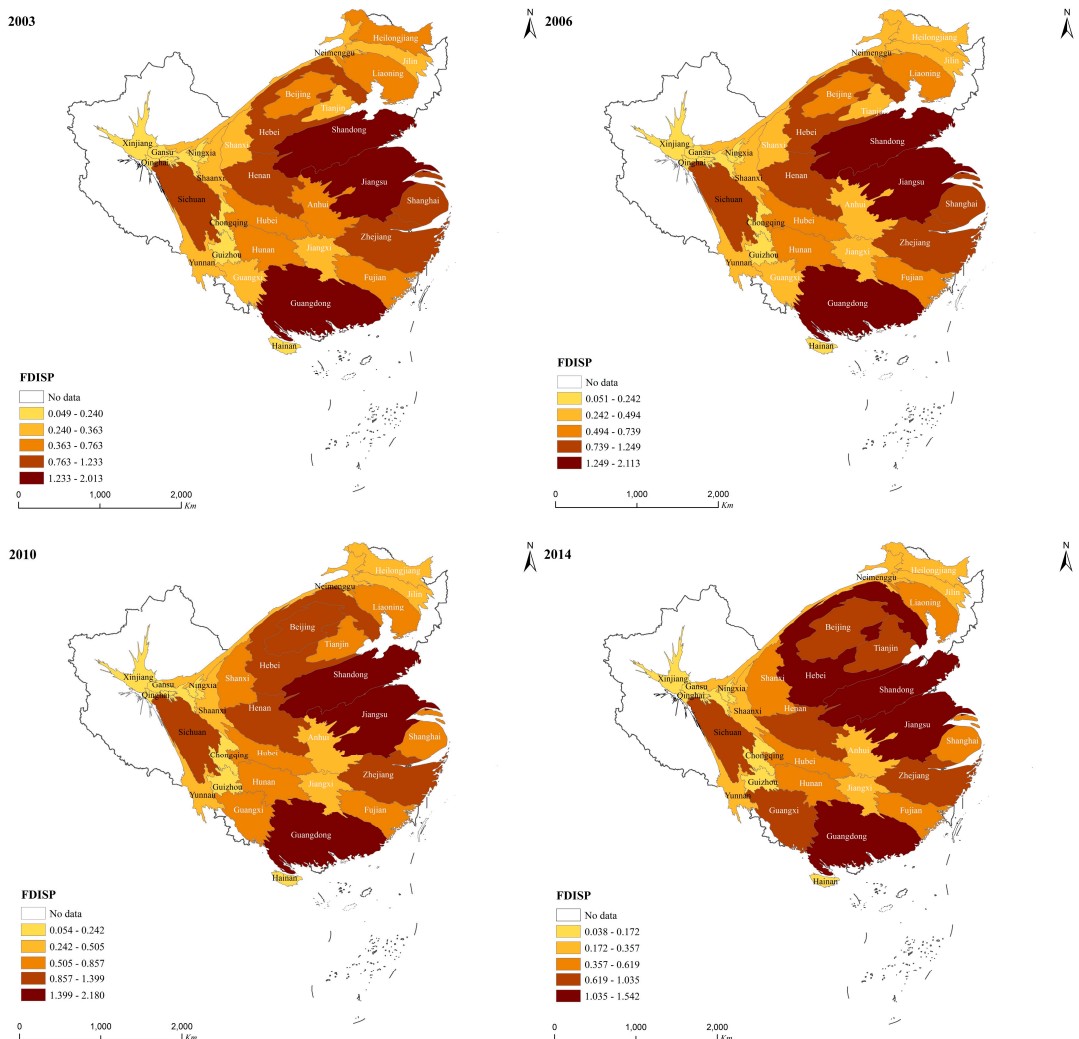

**Figure 4.** The spatial dynamics of China's technology spillover effect from FDI.

## 6. The Main Drivers of China's Technology Spillover Effect

### 6.1. Variables and Data

This paper uses the spatial panel model to verify the drivers of the space-time evolution of international technology spillover effect. Many factors influence the effect, including the investment willingness of the technology transfer subject (such as multinational corporations), and basic conditions (such as trade policy and the economic development of companies accepting assistance). Technology transfers tend take place in areas with good economic development, available financing, more open trade policies, more standardized management systems, a professional workforce, and an available supply of labor. This paper incorporates the related literature [15] and the availability of data to screen out five factors: Financial development, human capital, economic development, trade openness, and institutional factors. Variable meanings and data sources are shown in Table 1.

**Table 1.** Variable Meanings and Data Sources.

| Indicators | Indicator Meaning | Unit | Specific Indicators | Data Sources |
|:---:|:---:|:---:|:---:|:---:|
| **FD** | Financial development | % | Financial institution loan balance as a proportion of GDP | China Statistical Yearbook |
| **HC** | Human capital | Person | Student enrollment in regular institutions of higher education | China Statistical Yearbook |
| **ED** | Economic development | Yuan | Per capita GDP | China Statistical Yearbook |
| **TO** | Trade openness | % | Total import and export as a proportion of GDP | China Statistical Yearbook |
| **IF** | Institutional factors | % | Private economy added value as a proportion of GDP | China Statistical Yearbook |

*6.2. Regression Results and Analysis*

6.2.1. Regression Results of Import Trade Channel

On the basis of the aforementioned results, the default choice for an SLM (Spatial Lag Model) or SEM (Spatial Error Model), when estimating the spatial panel model, combines with temporal and fixed effects. Table 2 lists the regression results of the import trade channel. Comparing the *LogL* test value of the SLM and SEM models, because SLM's *LogL* was greater than SEM's *LogL*, the SLM model's effect was better [20–22]. According to the SLM model, from 2003–2014, the parameter estimations of human capital, economic development, trade openness and institutional factors were significant, except financial development. Each of them promoted the technological spillover effect of import trade channels positively, but the nature and intensity of the effect varied with time. Among them, the parameter estimations of human capital in 2003 and 2006 were not significant, but it passed the significant level test of 5% in 2010 and 2014, with a gradually increasing estimation coefficient. This shows that the impact of the increase in the technology absorption capacity brought about by the human capital improvement on the technology spillover effect of the import channel becomes more important over time. The parameter estimations of economic development and institutional factors were positive, and passed the significant level test of 1%. This shows that the economic development and institutional factors have promoted the technological spillover effect of import trade channels to some extent. From 2006 to 2010, although the parameter estimation of trade openness factor decreased from 1.229 to 0.530, it was relatively large compared with other drivers. This may be due to the impact of the subprime mortgage crisis in 2008.

**Table 2.** Regression Results of the Import Trade Channel.

| Variables | 2003 | | 2006 | | 2010 | | 2014 | |
|:---:|:---:|:---:|:---:|:---:|:---:|:---:|:---:|:---:|
| | SLM | SEM | SLM | SEM | SLM | SEM | SLM | SEM |
| FD | 0.042 (0.9910) | 0.037 (0.8813) | 0.072 (1.3273) | 0.078 (1.4022) | 0.057 (1.1121) | 0.043 (1.0028) | 0.083 (1.4932) | 0.095 (1.7885) *** |
| HC | 0.069 (1.028) | 0.072 (1.1130) | 0.083 (1.6239) | 0.082 (1.6014) | 0.225 (2.0078) ** | 0.276 (1.7294) *** | 0.292 (2.6432) ** | 0.343 (2.4801) ** |
| ED | 1.190 (3.9772) * | 1.083 (3.7484) * | 1.208 (4.3925) * | 1.594 (4.5333) * | 0.459 (3.6893) * | 1.229 (4.0155) * | 1.192 (3.4291) * | 1.286 (3.7208) * |
| TO | 1.125 (5.211) * | 1.323 (4.8397) * | 1.229 (3.3892) * | 1.942 (3.7820) * | 0.530 (4.2939) * | 0.437 (3.5024) * | 1.416 (3.8403) * | 1.635 (3.9215) * |
| IF | 0.723 (5.2511) * | 1.004 (6.2590) * | 0.997 (4.8023) * | 0.998 (4.8037) * | 0.402 (3.5893) * | 0.787 (4.289) * | 0.883 (4.3903) * | 0.983 (5.2302) * |
| LogL | 90.25 | 89.99 | 85.78 | 85.52 | 78.62 | 78.60 | 70.21 | 70.04 |

Note: *, **, *** indicate significant at the 1%, 5%, and 10% levels, respectively, the numbers in brackets indicate the *t* statistic value.

### 6.2.2. Regression Results of FDI channel

Table 3 lists the regression results of the import trade channel. Similar to the regression results of import trade channels, the SLM model was also selected for the regression results of FDI channel, based on the *LogL* test value. According to the SLM model, from 2003 to 2014, the parameter estimations of financial development, human capital, economic development, and institutional factors were significant, except trade openness. Each of them promoted the technological spillover effect of FDI channels positively, but the nature and intensity of the effect also varied with time. Among them, the increasing trend of the driving role of financial development was the most significant. This indicates that as China's financial system improves, the technology spillover effect of FDI channels has been further promoted. Human capital passed the significance test in 2003, 2006 and 2010, but failed to pass the significance test in 2014. This indicates that the influence of this factor on the technological spillover effect of advancing FDI channels changes from significant to random over time. The driving force of economic development had a downward trend in 2014, which may be due to the decline in the attractiveness of FDI after China's economy entered the new normal. Institutional factors showed insignificant results in 2003, but it passed the significant level test of 5% in 2006–2014, with a gradually increasing estimation coefficient. This indicates that the role of institutional factors in promoting the spillover effect of FDI channels has been verified.

**Table 3.** Regression Results of the FDI Channel.

| Variables | 2003 | | 2006 | | 2010 | | 2014 | |
|---|---|---|---|---|---|---|---|---|
| | SLM | SEM | SLM | SEM | SLM | SEM | SLM | SEM |
| FD | 0.421 (2.0840) ** | 0.348 (1.8233) *** | 0.525 (2.4465) ** | 0.662 (2.3795) * | 0.928 (4.2930) * | 0.923 (4.255) * | 1.390 (5.8023) * | 1.232 (5.4518) * |
| HC | 0.383 (2.7832) ** | 0.433 (2.3226) ** | 0.575 (3.2832) * | 0.682 (3.6305) * | 0.237 (1.9832) *** | 0.165 (1.6723) | 0.082 (1.5323) | 0.078 (1.4927) |
| ED | 0.783 (3.7652) * | 0.709 (4.2039) * | 0.830 (3.2767) * | 0.858 (3.3607) * | 0.873 (2.9978) * | 0.902 (3.0120) * | 0.829 (3.2758) * | 0.834 (3.6427) * |
| TO | 0.028 (0.9735) | 0.033 (1.0207) | 0.029 (0.9659) | 0.036 (0.8395) | −0.022 (−1.2039) | −0.028 (−1.0780) | −0.053 (−1.3927) | −0.048 (−1.3574) |
| IF | 0.111 (1.6392) | 0.309 (1.8205) *** | 0.307 (2.0456) ** | 0.427 (2.3890) ** | 0.419 (2.2612) ** | 0.440 (2.7639) ** | 0.374 (2.1092) ** | 0.398 (2.2308) ** |
| LogL | 82.66 | 82.39 | 83.87 | 83.40 | 79.23 | 79.34 | 73.64 | 73.53 |

Note: *, **, *** indicate significant at the 1%, 5%, and 10% levels, respectively, the numbers in brackets indicate the *t* statistic value.

### 6.2.3. Endogeneity Test of the Models

Because both IMSP and FDISP mutually reinforce with TO (Trade Openness), we must consider the endogeneity problems when studying the driving mechanism of FDI and import trade channels. Since it is impossible to find a suitable tool variable, we used the first-order delay of TO as the tool variable to estimate. By comparing the models' estimated coefficients, we found that positive and negative signs are consistent. The Durbin–Wu–Hausman (DWH) test showed that the *P* values are 0.1028 and 0.1217. Therefore, we do not reject the original assumption that the variables are endogeneity at the 10% significant level. This illustrated that the models' endogeneity problem is not serious, and that the empirical results of this paper do not have strong endogeneity problems.

## 7. Conclusions and Policies

### 7.1. Conclusions

As a whole, the stock of international technology spillovers in general has shown an upward trend in China, except in 2008 and 2012 which saw deep-V plunges. The plunge in 2008 was more dramatic. After 2011, the growth rate of international technology spillover from international trade and FDI respects slowed down obviously. With the drastic improvement of economic power, the technology innovation ability of China has been greatly enhanced. For this reason, the disparity between China and other developed countries has drastically narrowed, which means that the international technology spillovers into China gradually decreased. Therefore, its stock gradually tends to be flat.

In terms of the spatial pattern, the distribution of the international technology spillover from FDI was roughly similar to that from import trade. The regions of high-level effect were distributed in Guangdong, Zhejiang, and Shandong, the medium located in the central area and the low were in the Western area. What is more, the agglomeration effect was common to them. The provinces that are spatially distributed around Shandong, Guangdong, and Zhejiang provinces are mainly in the mid-level, which shows that agglomeration effects have emerged in the region. Meanwhile, based on the regional GDP and per capita GDP from 2003 to 2014, Beijing and Tianjin ranked in the top three in the regional GDP, although they were at the low level of the international technology spillover effect from FDI and import trade, which is quite contrary to the common belief that high level of international technology spillover represents a higher level of technology innovation capability, higher rate of economic growth, and naturally higher GDP and GDP per capita.

From the driving mechanism, human capital, economic development, trade openness and institutional factors promoted the technological spillover effect of import trade channels positively, and financial development, human capital, economic development, and institutional factors promoted the technological spillover effect of FDI channels positively, but the nature and intensity of the effect of them varied with time.

### 7.2. Policies

Generally speaking, the effect of the international technology spillover from import trade and FDI is relatively low in the regions that have low GDP and per capita GDP, except Beijing and Tianjin. Hence, the government is expected to pay attention to the introduction of foreign capital into the central and western area. This is because these regions which suffer disadvantages in the competition in various aspects, including capital, environment and human capital, are not among the optimal choices of multinational corporations when they choose investment destinations. In order to change this situation, it is the central government that can make the adjustment. The unbalanced development of regional economy and competitions among the regions with each province has all levels of government and has their own development goals. Under such circumstance, the national government is the best regulator to change it among different regions.

The agglomeration effect of the international technology spillovers from FDI emerges in certain regions. Specifically speaking, the level of the international technology spillover effect from FDI around that of high regions is relatively high, which means developed regions are able to boost the development of other regions around it. For this reason, the Western area needs a key city or region to propel its development. Of course, due to the limited driving force and several competitive factors between the West, and the central, and the East, fundamentally speaking, the development of the Western area should depend on itself.

The international technology spillover has steadily increased since 2011. China still lag behind the developed countries even though it has made considerable achievements in technology and economy. As the developed countries and regions are important sources of China's international technology spillovers, China should pay attention to the introduction of FDI and improve the domestic investment environment.

**Author Contributions:** C.L. developed the original idea of this study, revised the manuscript and performed the data analysis. Q.G. supervised the research project and drafted the original manuscript. All authors read and approved the final manuscript.

**Funding:** This research was funded by the National Nature Science Foundation of China, grant number 41571123, the National Social Science Fund of China, grant number 18BJL056, the Shanghai Pujiang Program, grant number 17PJC030 and the Strategic Priority Research Program of the Chinese Academy of Sciences, grant number XDA20100311.

**Acknowledgments:** We are grateful for Tao Wang portraying the figures.

**Conflicts of Interest:** The authors declare no conflict of interest.

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
