# Peer review of "Technology Spillover Effect in China: The Spatiotemporal Evolution and Its Drivers"

_sustainability, doi:10.3390/su11061694_

Round 1
Reviewer 1 Report
From the perspective of spatial-temporal evolution, this paper studies the effects of foreign technology spillovers on China in different periods and regions, which is quite innovative. However, the manuscript has the following problems: 1)In the section “5.1. Indicator Selection”, What are the criteria for index selection? Why are these indicators chosen? These need to be explained in detail 2)Introduction, the first study too little, please enrich! At the same time, the literature listed traces obvious, lack of logic. 3)I found another similar literature, please refer to and quote. “The Influencing Factors, Regional Difference and Temporal Variation of Industrial Technology Innovation: Evidence with the FOA-GRNN Model. Sustainability, 2018, 10(1): 187.” 4)How is space distance defined? Please explain and mark.Author Response
From the perspective of spatial-temporal evolution, this paper studies the effects of foreign technology spillovers on China in different periods and regions, which is quite innovative. However, the manuscript has the following problems:
1.In the section “5.1. Indicator Selection”, what are the criteria for index selection? Why are these indicators chosen? These need to be explained in detail.
Response: Thank you so much. It is an enlightening comment. We have explained the reasons for choosing these indicators.There are many factors affecting the international technology spillover effect, which generally include two aspects, one is the investment willingness of the technology transfer subject (such as multinational corporations), and the others are the basic conditions, trade policy and economic development environment of technology acceptance subject. The main body of technology transfer tends to invest those areas with good economic development, good financing environment, more open trade policies, more standardized management system and rich professionals and labor. Therefore, from the above aspects, this paper combines the related literature and the availability of data to screen out five factors: financial development, human capital, economic development, trade openness and institutional factor. (Please see the text of line 397-407 in section 6.1)
2.Introduction, the first study too little, please enrich! At the same time, the literature listed traces obvious, lack of logic.
Response: Many thanks. According to the advice, we have supplemented and revised the introduction, and rewritten the literature review as a separate part. Correspondingly, we adjusted the references. (Please see the section 1 and section 2)
3.I found another similar literature, please refer to and quote. “The Influencing Factors, Regional Difference and Temporal Variation of Industrial Technology Innovation: Evidence with the FOA-GRNN Model. Sustainability, 2018, 10(1): 187.”
Response: Thanks for your kind reminder. According to the comment, we havequoted this reference in the process of analysis. (Please see the text of line 405 in section 6.1)
4.How is space distance defined? Please explain and mark.
Response: Thanks for your kind reminder. We added the space distance scale to the maps and explained the spatial scale in the text. (Please see figure3 and figure 4)
Reviewer 2 Report
a. I suspect that in the results obtained (see tables 2 and 3), they have the following methodological problems:
i. Multicollinearity To verify the absence of correlation between the exogenous, the authors must include (as an annex) the matrix of correlations.
ii. I suspect that the results present problems of endogeneity. The authors should be analyzed in the manuscript (see Girma 2005; Nowak-Lehmann et al. 2012; Anward and Nguyen 2014). Applied Durbin–Wu–Hausman test (DWH) or regressor endogeneity Test to check if it has achieved to avoid the bias caused by endogeneity problems.
iii. It would be very important to calculate a learning capacity threshold since the difference between the different regions can be significant. The authors should detail how he has found the thresholds for his model. See Girma (2005) and Ubeda and Perez-Hernandez (2016).
Author Response
I suspect that in the results obtained (see tables 2 and 3), they have the following methodological problems:
1. Multicollinearity to verify the absence of correlation between the exogenous, the authors must include (as an annex) the matrix of correlations.
Response: Thank you so much. We have modified the article in accordance with this recommendation. To verify the absence of correlation between the exogenous, the matrix of correlations, is shown in Annex 1. It can be seen that except for the correlation coefficient between ED and FD variables slightly more than 0.3, the other variables are all less than 0.3. Therefore, we can consider that the variables passed the multicollinearity test. (Please see the Annex 1)
2. I suspect that the results present problems of endogeneity. The authors should be analyzed in the manuscript (see Girma 2005; Nowak-Lehmann et al. 2012; Anward and Nguyen 2014). Applied Durbin–Wu–Hausman test (DWH) or regressor endogeneity Test to check if it has achieved to avoid the bias caused by endogeneity problems.
Response: Thank you so much. According to the advice, we have added the Durbin–Wu–Hausman test toavoid the bias caused by endogeneity problems. (Please see the text of line 466-475 in section 6.2.3)
3. It would be very important to calculate a learning capacity threshold since the difference between the different regions can be significant. The authors should detail how he has found the thresholds for his model. See Girma (2005) and Ubeda and Perez-Hernandez (2016).
Response: Thank you so much. This paper only analyzed the temporal and spatial evolution of international technology spillover effects obtained from China's import trade channels and FDI channels. It does not cover the study of a learning capacity threshold. This point has been discussed in the literature review (section 2). And we absolutely believe that the learning capacity threshold will be one of the important directions in our future research.
Reviewer 3 Report
The topic of the paper is relevant and interesting. In the following part are some suggestions for revising some methodological explanations.
In the part where evolutionary technology spillovers from international import trade and FDI are presented, it is suggested that the figures 3 and 4 are better explained. The reader can not simply conclude what do different colors mean, and numbers in the legend are not clear. Also, in the part 5.2. Regression Results and Analysis - it is suggested that author/s disscuss why did they choose SLM model and name few studies that used the same model, so the logic of the research is clear.
For further research it is suggested to author/s to estimate the effect of international technology spillover with considering the export trade as well.
Author Response
The topic of the paper is relevant and interesting. In the following part are some suggestions for revising some methodological explanations.
1. In the part where evolutionary technology spillovers from international import trade and FDI are presented, it is suggested that the figures 3 and 4 are better explained. The reader cannot simply conclude what do different colors mean, and numbers in the legend are not clear.
Response: Thanks for your kind advice. According to the comment, we have modified the figures to mark the meaning of the different colors and increase the size of the number in the legend.
2. Also, in the part 5.2. Regression Results and Analysis - it is suggested that author/s disscuss why did they choose SLM model and name few studies that used the same model, so the logic of the research is clear.
Response: Many thanks. We have modified the article in accordance with this recommendation. By comparing the LogL test value of SLM and SEM model, SLM’s LogL was greater than SEM’s LogL, so the effect of SLM model was relatively better,and we quoted three studies(Liu and Zou(2012),Smith et al.(2016) and Blasques et al.(2016)) that used the same model. (Please see the text of line 424-428 in section 6.2.1)
3. For further research it is suggested to author/s to estimate the effect of international technology spillover with considering the export trade as well.
Response: Thank you so much. It is an enlightening comment. We will estimate the effect of international technology spillover with considering the export tradein our future research.
Reviewer 4 Report
This paper analyzes the effect of international technology spillover from FDI and international trade in China during the period of 2003 to 2014. The article is well written and the issue is interesting.
I have appreciate the comprehensive picture of the state of the art of this interesting subject, even if it is scattered in various sections in the present version. So I suggest to include a separate section "literature review".
I have some reservations about the presentation of results: 1) the econometric model and the estimator used are not well explained, 2) tables of results are hard to understand (there are too many acronyms); 3) does the panel model include time and fixed or random effects?
Finally, in my opinion the author(s) should better highlight the contribution of this work to the existing literature and explain why it is in line with the aim and scope of Sustainability.
Author Response
This paper analyzes the effect of international technology spillover from FDI and international trade in China during the period of 2003 to 2014. The article is well written and the issue is interesting.
1. I have appreciate the comprehensive picture of the state of the art of this interesting subject, even if it is scattered in various sections in the present version. So I suggest to include a separate section "literature review".
Response: Thanks for your kind reminder. According to the advice, we have supplemented and revised the introduction, and rewritten the literature review as a separate part. (Please see the section 1 and section 2)
2.I have some reservations about the presentation of results: 1) the econometric model and the estimator used are not well explained, 2) tables of results are hard to understand (there are too many acronyms); 3) does the panel model include time and fixed or random effects?
Response: Many thanks. We have modified the article in accordance with this recommendation. 1) We have supplemented the explanation of the econometric model and the estimator in the text (Please see the section 6.1 and section 6.2). 2) The meaning of acronyms in tables (table 1, table 2 and table 3) can be found in the second column (Indicator meaning) of table 1. 3)Through the analysis of the spatial-temporal evolution of the international technology spillover effect, the default choice for SLM(Spatial Lag Model) or SEM(Spatial Error Model),when estimating the spatial panel model, was both with both temporal and fixed effects.
3. Finally, in my opinion the author(s) should better highlight the contribution of this work to the existing literature and explain why it is in line with the aim and scope of Sustainability.
Response: Thanks. We have modified the article in accordance with this recommendation.Most of existing studies have confirmed that import trade and FDI are two important channels of international technology spillover effect, but few scholars analyze the spatial-temporal evolution of the international technology spillover effect, this paper respectively discusses the spatial-temporal evolution of the international technology spillover effect from import trade and FDI channels and its driving mechanism, and policy recommendations based on spatial-temporal evolution and its driving mechanism are more targeted, which will help China to make better utilization of technology spillover effect.
In addition, it can guide developing countries to undertake and absorb technological spillovers from developed countries through the study of the temporal and spatial evolution of international technology spillover effects in this paper. And thereby will enhance the technological innovation capabilities of developing countries and their position in global value chains, promote the transformation of traditional development modes and achieve sustainable economic and social development.
(Please see the text of line 172-185 on the end of section 2)
Round 2
Reviewer 1 Report
The article has been greatly revised and I think it can be accepted and published.